# Physical Activity Friendliness of Neighborhoods: Do Subjective and Objective Measures Correspond Within a Mid-Sized Dutch Town?

**DOI:** 10.3390/ijerph22040536

**Published:** 2025-04-01

**Authors:** Thomas G. Kuijpers, H. Susan J. Picavet, Jeroen Lakerveld, Johannes Mark Noordzij, G.C. Wanda Wendel-Vos, Barbara A. M. Snoeker

**Affiliations:** 1Center for Prevention, Lifestyle and Health, National Institute of Public Health and the Environment, 3721 MA Bilthoven, The Netherlands; thomaskuijpers@gmail.com (T.G.K.); susan.picavet@rivm.nl (H.S.J.P.); wanda.vos@rivm.nl (G.C.W.W.-V.); 2Department of Epidemiology and Data Science, Amsterdam Public Health Research Institute, Amsterdam UMC, Vrije Universiteit Amsterdam, 1081 HV Amsterdam, The Netherlands; j.lakerveld@amsterdamumc.nl; 3Mulier Institute, 3584 AA Utrecht, The Netherlands; m.noordzij@mulierinstituut.nl

**Keywords:** physical activity, geospatial measures, perception, epidemiology, neighborhoods

## Abstract

One potential strategy to promote physical activity (PA) involves changing the neighborhood environment. The PA-friendliness of neighborhood environments is often calculated using geospatial data. However, the association with perceived PA-friendliness seems to be low. Therefore, we examined the relationship between two Dutch geospatial measures and residents’ perceptions regarding the PA-friendliness of their neighborhoods. Data from 3438 respondents aged 40–80 years from the Doetinchem Cohort Study were linked to individual geospatial data. In addition to respondents’ self-reports, we used the following two geospatial measures: the Dutch walkability index and the Dutch indicator for PA-friendly environments (KBO-indicator). We performed logistic regression analyses to assess associations between perceived PA-friendliness and two objective geospatial measures, including interactions for age, sex, education, work status, and physical functioning. The majority of respondents (83%) evaluated their neighborhood as PA-friendly. The logistic analyses revealed no associations between the geospatial measures of walkability and PA-friendliness and the individuals’ perception measures. Similarly, a comparison of residents from high versus low walkable or PA-friendly neighborhoods demonstrated no significant differences in their perception of PA-friendliness. Additionally, no significant interaction effects were observed with sex, age, education, employment status, or physical functioning, indicating that even among subgroups there was no correspondence between objective and subjective measures. The lack of correspondence between both objective and subjective measures for PA-friendliness in the neighborhood suggests that these are distinct constructs. Future research should focus on qualitative methods to bridge the gap between objective and subjective measures and test whether the perceived PA-friendliness is similar to the objective measures.

## 1. Introduction

In Europe, about one-third of adults engage in low levels of physical activity (PA), with substantial differences across countries [1]. One possible explanation for these differences might be in the design of the built environment [1]. Ecological models, such as Bronfenbrenner’s [2], emphasize that health behaviors are shaped by influences across multiple levels, including the intrapersonal, interpersonal, organizational, community, and public policy levels [3]. Therefore, a PA-friendly environment can have an important influence on health behavior.

One potential strategy to promote PA involves changing the neighborhood environment to encourage PA among residents. This approach is endorsed by both the World Health Organization and the Dutch government [1,4]. Examples of environmental features of influence on PA are easy access to amenities by foot or bicycle, safe and attractive cycling, and walking infrastructures, and availability of attractive green or public spaces for physical activities [4,5]. In formulating strategies to change the neighborhood environment, policymakers, researchers, or urban planners may be informed by geospatial measures, often based on registered data. These can be used to characterize, benchmark, and monitor PA-friendly environments, for example, between different municipalities [6,7]. In the Netherlands, two measures to grasp the PA-friendliness of the environment were developed in 2016: the Dutch walkability index and the Dutch indicator for PA-friendly Environments (abbreviated in Dutch as the KBO indicator; “Kernindicator Beweegvriendelijke Omgeving”) [6]. The walkability index specifically captures aspects of the environment that encourage walking, while the KBO indicator measures various elements that are expected to facilitate and/or stimulate PA and sports participation both for specific activities and in general [6,7].

Next to measures using ‘objective’ geospatial data, also self-reported (i.e., subjective) measures are used to measure the PA-friendliness of the environment. These may consist of a few questions on perception of the environment or a complete scale such as the Neighborhood Environment Walkability Scale (NEWS) questionnaire [8]. Both subjective and objective measures of PA-friendliness of environments can include a wide range of elements, such as sidewalk surface area, green spaces, land use diversity, or proximity to amenities and retail destinations.

The relationship between subjective and objective measures of the PA-friendliness of the environment is a topic of many publications [9,10,11,12,13,14,15,16,17,18,19,20,21,22]. These investigations have yielded inconsistent results, largely due to variations in settings, research populations, methodologies, and environmental aspects being examined. However, they generally report low to moderate correspondence between objective and subjective environmental attributes. In a recent systematic review, authors suggest that objective and subjective measures may capture different aspects of the environment and may account for unique variance in PA behaviors [22]. Most of the available studies are carried out outside Europe [22]. A few studies have been executed within Europe, but the Netherlands is particularly interesting because of its socio-spatial infrastructure with densely populated areas and many cycle paths [23]. Furthermore, it has both cities and rural areas, and compared to other countries, there is much publicly available geospatial data with high-quality data, accessibility, and infrastructure, which enables comprehensive environmental analyses of the PA-friendliness of different neighborhoods [24]. The only study conducted in the Netherlands investigated the correspondence between subjective and objective measures of the availability of parks, sports facilities, sidewalks, and bicycle lanes among 654 adolescents in Rotterdam [25]. This study found a very poor level of agreement between both measures [25]. We aim to build on this research and therefore evaluate the correspondence between objective and subjective indicators of a PA-friendly environment in a different setting—a rural mid-sized town in the east of the Netherlands—with different objective and subjective measures and an older population. We hypothesize that objective and subjective measures correspond partially.

## 2. Materials and Methods

### 2.1. Study Population

For the individual-level data, we used data from the Doetinchem Cohort Study. This study provides data on biological and lifestyle factors to study their influence on health throughout the life course [26,27]. Every five years since 1987 (round 1), respondents complete questionnaires and undergo medical examinations. Initially, all respondents lived in Doetinchem, a mid-sized rural town in the eastern part of the Netherlands. The study was conducted according to the principles of the World Medical Association Declaration of Helsinki and its amendments since 1964 and in accordance with the Medical Research Involving Human Subject Act (WMO). The study protocols were approved by the Medical Ethics Committee of the University Medical Center Utrecht, and all informed consent was obtained from all participants [26,27]. For the present study, we used cross-sectional data from round six, collected in the years 2013–2017. The data included personal characteristics and the participants’ perception of a PA-friendly environment. The initial sample in round 1 was an age-sex stratified random sample of the population. All participants who were still alive, were still living in Doetinchem (had not emigrated), and had not actively withdrawn from the study were invited for the sixth measurement round. For the sixth round, 4455 respondents were invited, and 3438 participated, yielding a response rate of 77%.

### 2.2. Variables

Age, sex, education, working status, and physical functioning were added as covariates. Participants from the DCS were between 40 and 89 years old. Education included low (intermediate secondary education or less), intermediate (intermediate vocational and higher secondary education), and high education (higher vocational education or university). Work status indicates an individual as employed or as unemployed. If a person was retired, this was also categorized as unemployed. Physical functioning is measured using the subscale of the Short Form 36 Health Survey [28]. This scale assesses an individual’s ability to perform various physical activities and tasks. It encompasses aspects such as mobility, self-care, and everyday activities [28]. Scores on this subscale range from 0 to 100, with higher scores indicating better physical functioning and lower scores indicating worse physical functioning. We presented scores in quartiles (0–25, 25–50, 50–75, 75–100).

### 2.3. Geodata Variables Measuring the PA-Friendliness of the Neighborhood

In this study, we used two recently developed national-level geospatial measures to evaluate the walkability and PA-friendliness of the environment: the Dutch walkability index and the KBO indicator.

The Dutch walkability index was used on a neighborhood level and three buffer zones from the address of individual respondents (250 m, 500 m, 1000 m). This index is a composite score constructed by the Geoscience and Health Cohort Consortium (GECCO), based on seven components selected from previous walkability studies deemed influential for spatial variation in the Netherlands [29,30]. The spatial components were calculated as values assigned to raster cells within a uniformly spaced grid that covers the Netherlands, with each cell measuring 25 by 25 m [29]. The 25 by 25 m grid covers the entire country and is not necessarily a subset of the neighborhood scores or buffer zones. Rather, neighborhood scores and buffer zones can encompass multiple raster cells. The walkability index scores for entire neighborhoods and various buffer radii from respondents’ individual addresses were then aggregated from these grid cells. The components are as follows: (1) Population density, (2) Retail and service density, (3) Land-use mix (the heterogeneity of land use), (4) Street connectivity or intersection density (ratio between the number of true intersections (three or more legs) to the size of the selected area), (5) Green spaces (parks and gardens, forests, cemeteries), (6) Sidewalk surface area, and (7) Public transport density [4,6]. Each component was standardized (z-scores), averaged, and scaled between 0 and 100, with higher scores indicating greater walkability. Areas with a higher walkability score reflect a higher population density, better-connected street networks, a more diverse mix of land uses (e.g., residential, commercial, and recreational), greater availability of green spaces, more extensive sidewalk coverage, and improved accessibility to public transport [28]. Scores range from 0 to 100, with 0 indicating the lowest score (not walkable) and 100 indicating the highest score (completely walkable). For a comprehensive discussion on the rationale and detailed calculations of this measure and its components, see Wagtendonk and Lakerveld (2019) [29].

The KBO indicator was defined as “the extent to which the public physical environment provides opportunities for people to participate in sports and PA”. This indicator consists of four components: (1) The diversity and proximity of sports facilities such as football fields, hockey fields, tennis courts, sports halls, fitness centers, and swimming pools, (2) The proximity of sports and play areas in the public domain, (3) The proximity and surface area of recreational green and blue spaces, and (4) Proximity to amenities (including average distance to schools or supermarkets). The KBO indicator is calculated as the average of the scores for the available sub-indicators, each ranging from 0 to 100. Areas with a higher KBO-indicator score reflect better proximity and greater variety of sports facilities (e.g., football fields, tennis courts, sports halls, and swimming pools), more accessible public sports and play areas, greater availability of recreational green and blue spaces (e.g., parks and water bodies for recreation), and shorter distances to amenities such as supermarkets and schools. Higher scores thus indicate improved conditions for PA [30]. Scores range from 0–100, with 0 indicating the lowest score (not PA-friendly) and 100 indicating the highest score (completely PA-friendly). Detailed descriptions and calculations can be found on the KBO methods webpage [31].

### 2.4. Perception of PA-Friendly Environment

To assess how respondents perceive the PA-friendliness of their neighborhood environment for PA, they were asked whether they ‘perceived any circumstances in their neighborhood that make it less appealing to engage in sports and physical activity’ (yes or no).

### 2.5. Statistical Analyses

To investigate the relationships between the two geospatial measures and our outcome measure of the perceived PA-friendliness of neighborhoods, we conducted logistic regression analyses including age category, sex, educational level, employment status, and physical functioning as covariates. Geospatial measures were treated as individual-level attributes. Additionally, interaction effects with the covariates were assessed. For the walkability index, we tested three buffer zones (250 m, 500 m, and 1000 m from the respondent’s home address) (see Table A1 in Appendix A). Furthermore, we conducted multilevel sensitivity analyses with a random intercept for neighborhoods to account for the hierarchical structure of the neighborhood-level data, as there were up to 186 respondents residing in the same neighborhood (see Appendix A, Table A2 and Table A3).

To examine whether individuals in high (≥Quartile 3) walkable or PA-friendly environments perceived their environment as more PA-friendly compared to individuals in low (Quartile 1) walkable or PA-friendly environments, we conducted *t*-tests. The classification into high and low groups was data-driven, based on the available scores, as no theoretical criteria existed for this distinction. Consequently, individuals were grouped according to their relative position within the score distribution.

To assess agreement between subjective and geospatial measures, kappa scores were calculated between the walkability index, KBO indicator, and the perception measure. Dichotomous measures of the geospatial variables were created based on quartiles: Q1 + Q2 were classified as ‘not PA-friendly’ and Q3 + Q4 as ‘PA-friendly’. Kappa between the two geospatial measures used four categories: Q1, Q2, Q3, and Q4.

Lastly, *t*-tests were conducted to explore whether Doetinchem neighborhoods score differently from other Dutch neighborhoods on the walkability index scores or KBO indicator scores.

All analyses were performed using R version 4.1.2. Geodata values and neighborhood delineations were chosen for the year 2019, as this was the closest match to the individual-level data collection period (2013–2017).

## 3. Results

### 3.1. Descriptives

Slightly over half of the study population (53.2%) were women, and 43.5% reported a low educational level. The population was quite healthy, with 74.8% having a relatively high level of functioning (75–100 points). The majority of participants (83.6%) did not report any conditions in their neighborhoods that made it less appealing to engage in sports and PA (Table 1).

Since 1987, 385 participants (11%) relocated to other regions of the Netherlands, primarily to neighboring municipalities. The remaining 3048 participants (89%) continued to live in Doetinchem and are included in this study. Included respondents are distributed across 272 individual neighborhoods, of which 73 are within the municipality of Doetinchem. The number of participants per analyzed neighborhood ranges from 1 to 186, with a mean of 60 and a median of 33. See Table A4 in Appendix A for an overview of the number of participants per neighborhood. Figure 1 displays the neighborhoods in the town of Doetinchem, along with their corresponding values for the walkability index (2019) and KBO indicator (2020; based on 2019 data).

Figure 2 and Figure 3 present boxplots for the walkability index and the KBO indicator for both levels (yes, no) of perception of a PA-friendly environment. These figures demonstrate an almost complete overlap in variance between the two perception levels and both geospatial measures. Boxplots for the three buffer zone scores of the walkability index (250 m, 500 m, and 1000 m) can be found in Appendix A (Figure A1, Figure A2 and Figure A3).

### 3.2. Association Between Objective and Subjective Measures

After adjusting for covariates, there is no association between the objective and subjective measures of PA-friendliness, not for the walkability index (OR = 0.99, 95%CI = 0.99–1.01) nor the KBO indicator (OR = 1.00, 95%CI = 0.99–1.01).

### 3.3. Interaction and Sensitivity Analyses

Interaction effects with the covariates were tested and were all found to be insignificant (see Table A5 in Appendix A). The analyses for the buffer zones (250 m, 500 m, and 1000 m) of the walkability index provided similar results as the main analyses with the neighborhood scores (see Table A1). Multilevel analyses of neighborhood-level values for the walkability index and KBO indicator, also presented in Appendix A (Table A2 and Table A3), similarly show no correspondence.

### 3.4. High vs. Low Walkability Index Compared to the Perception of PA-Friendly Neighborhoods

A comparison of individuals residing in low (≤Quartile 1) versus high (≥Quartile 3) walkability-friendly environments revealed no significant differences in their perceptions of a PA-friendly environment. In low-walkable neighborhoods (*n* = 877), 85% of individuals perceived their neighborhoods as PA-friendly, compared to 82% in high-walkable neighborhoods (*n* = 893); *t*(1587.5) = 1.4507, *p* = 0.1471. This was similar in neighborhoods that scored low (≤Quartile 1) versus high (≥Quartile 3) on PA-friendliness as measured by the KBO indicator (*n* = 888; 83% vs. 84% of the residents in those neighborhoods perceived their neighborhood to be PA-friendly, respectively).

### 3.5. Agreement Statistics

Weighted kappa scores indicated less than chance agreement between the perception variable and the dichotomized walkability index (κ = −0.04) and between the perception variable and the dichotomized KBO indicator (κ = −0.018). The weighted kappa score between the two geospatial variables was poor (based on four quartile levels) (κ = 0.012).

### 3.6. PA-Friendliness in Doetinchem vs. The Netherlands

Lastly, we compared the mean walkability index and KBO indicator scores for neighborhoods in Doetinchem with those of all Dutch neighborhoods. The independent samples *t*-tests showed that Doetinchem neighborhoods had significantly higher walkability index scores (*t*(3058) = 34.4, *p* < 0.001) but did not differ significantly in KBO-indicator scores (*t*(3058) = −85.3, *p* = 1.00). Boxplots illustrating the variance for Doetinchem versus all Dutch neighborhoods are available in Appendix A (Figure A4 and Figure A5).

## 4. Discussion

The current study found no associations between two Dutch geospatial measures of walkability and PA-friendliness and the subjective individuals’ perceptions of PA-friendliness. The lack of associations was consistent for neighborhood scores and three different buffer zones for walkability. Additionally, no significant interaction effects were observed with sex, age, education, employment status, or physical functioning, indicating that even among subgroups there was no correspondence between objective and subjective measures.

These findings are consistent with other studies described in a previously published systematic review [22]. The authors of that review suggested that objective and subjective assessments of the PA-friendliness of the environment likely capture different dimensions of the environment and contribute uniquely to various PA behaviors [22]. Moreover, other research suggests that while PA behavior is related to objective measures of PA-friendliness, this effect is likely mediated by individual perceptions of that same environment, i.e., the neighborhood environment shapes the perceptions of the people who reside in it [32]. Sallis and colleagues (2006) emphasized that perceptions of the environment are influenced by social, cognitive, and affective processes, while objective measures are more stable and less biased [33]. Different individuals can form different cognitive representations of the same neighborhood environment, but the process by which these perceptions are formed remains unclear. Orstad and colleagues conclude that most scholars investigate moderation by age, gender, education, PA level, body mass index (BMI), and perceptions of the neighborhood [22]. However, perhaps other (unmeasured) factors are more relevant in shaping the perceptions of the environment, such as cultural background, family situation (living alone or with kids), accessibility of PA-friendly areas, having a dog, or the extent of interaction with the environment. Future scholars should use more qualitative research designs to explore which factors are relevant in shaping the perceptions of the neighborhood environment.

Most studies investigating the agreement between objective and subjective PA-friendly environments use kappa statistics, which typically show poor to moderate agreement [34,35,36,37,38,39,40,41,42,43,44,45,46,47,48,49,50,51,52]. This may suggest that the relationship may not be straightforward and that, for example, there is a threshold effect. People might only evaluate their environments accurately when they are either very PA-friendly or very PA-unfriendly. For example, the chance that Dutch adolescents perceived parks as available was higher when more parks were present in their environment [24]. A study in Belgium found significant differences in perception between residents of high versus low-walkable neighborhoods [15]. However, even when comparing neighborhoods with high versus low walkability and PA-friendliness, we found no statistically significant association in our sample. One explanation could be that the Netherlands, and especially the area of Doetinchem, is relatively PA-friendly. Therefore, even the neighborhoods with lower scores on PA-friendliness are still rated as PA-friendly.

Although we also expected similarities between objective and subjective measures, our findings suggest that the individual perceptions of PA-friendliness are based on other constructs than the objective measures. This has also been suggested in a previous study [53]. One explanation for the differences from this study was that persons who were more active attributed these activities to their neighborhood environment, while those who were less active blamed their environment, though the environment was objectively similar. Another explanation was that the objective measures also captured spaces that were not all publicly accessible and therefore did not represent the actual qualities and participant usage. For example, objectively there were green spaces in the area so the neighborhood was rated as PA-friendly, while persons living in the neighborhood rated it as unfriendly as they were restricted to enter some of the green areas. Both explanations were relevant to our study. A final explanation is that individuals may perceive neighborhood boundaries differently, focusing on specific areas, which alters their perception of what encompasses one’s neighborhood. What individuals perceive as their neighborhood may differ per individual and may also be different from the register-based boundaries. It is quite difficult to distinguish between the influence of the objective environment on subjective perception and vice versa. Our examples reflect both situations.

In our study, although the objective measure rated a neighborhood as PA-friendly, the subjective measure showed no correlation with the objective one. Moreover, all measures, objective as well as subjective, measure different constructs. The walkability index is a measure that combines key spatial components with a focus on walkability, while the KBO indicator measures the extent to which the public space provides opportunities to participate in sports and PA. The KBO indicator is therefore more focused on sports facilities and play areas. Also, the KBO indicator presents an average score, and two completely different areas could therefore be similarly rated. For example, if an area has green spaces but no play areas, KBO indicator scores are the same as an area with play areas but without green spaces. The perception of both areas, however, can be very different.

The lack of correspondence between objective and subjective measures could hypothetically also be due to a ceiling effect, as Doetinchem could be more PA-friendly than other parts of the Netherlands. However, our additional analyses showed that while Doetinchem is not more PA-friendly according to the KBO indicator, it does score higher on walkability than other Dutch neighborhoods. To better understand this relationship, it would be advisable to test the walkability index in other regions of the Netherlands with more observations in the lower ranges of walkability scores.

Based on our results, we suggest that subjective measures give insight into the actual use of the neighborhood environment and serve another purpose than the objective measures. For policy purposes, it could, therefore, be interesting to not only focus on objective indexes. Objective measures can be used, although it should be tested whether these measures are similarly perceived subjectively. We suggest including qualitative studies to bridge the gap between objective and subjective perceptions of the PA-friendliness of the neighborhood environment and test whether the perceived PA-friendliness is similar to the objective measures.

Previous European studies focused on urbanized settings [34,42,43,52,53,54,55,56]. A strength of this study is that it is a new European study in a Dutch context, in a less urbanized setting with a relatively large sample size. We were able to assess two recently developed geospatial Dutch measures in this study.

An important limitation of this study is the use of a dichotomous perception variable. Although it has been described as a perception of PA-friendliness, it is not that comprehensive. Future research would benefit from employing a more nuanced and comprehensive measure, with a ten-point Likert scale, to capture a wider range of perceptions and introduce more variation in responses. Additionally, the original question’s double-negative framing may have confused respondents, potentially affecting the accuracy of their answers. Future Dutch researchers might consider using the NEWS questionnaire [8], which provides a more nuanced and valid measure of walkability, to explore its correlation with the Dutch walkability index. To measure residents’ perceptions of various aspects of PA friendliness in their neighborhood, beyond walkability, we recommend using the PANES (Physical Activity Neighborhood Environment Scale). This scale assesses 17 items related to the neighborhood’s safety, attractiveness, infrastructure, and access to different destinations [57].

Although we distinguish objective measures from subjective measures, objective measures also involve some degree of subjectivity. For example, if we consider the Walkability index, most components (for example, land-use mix, green spaces) cannot be obtained completely objectively. For example, the definition of “green space” may be a subject of debate.

## 5. Conclusions

Our research indicates that objective indicators do not correspond to our subjective indicator of a PA-friendly environment. These may capture different aspects of the PA-friendliness of the environment. Further research is needed to clarify what these constructs specifically measure. Policymakers should consider not only using objective measures of PA-friendliness but also utilizing qualitative methods to test whether the perceived PA-friendliness is similar to the objective measures.

## Figures and Tables

**Figure 1 ijerph-22-00536-f001:**
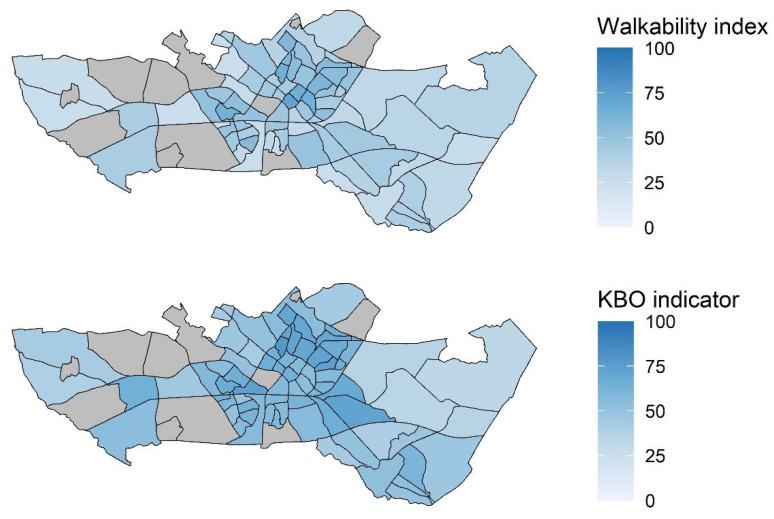
Neighborhoods in the municipality of Doetinchem and their corresponding walkability index (2019) and KBO indicator (2020) values (grey areas indicate missing data, or no respondents living in those neighborhoods).

**Figure 2 ijerph-22-00536-f002:**
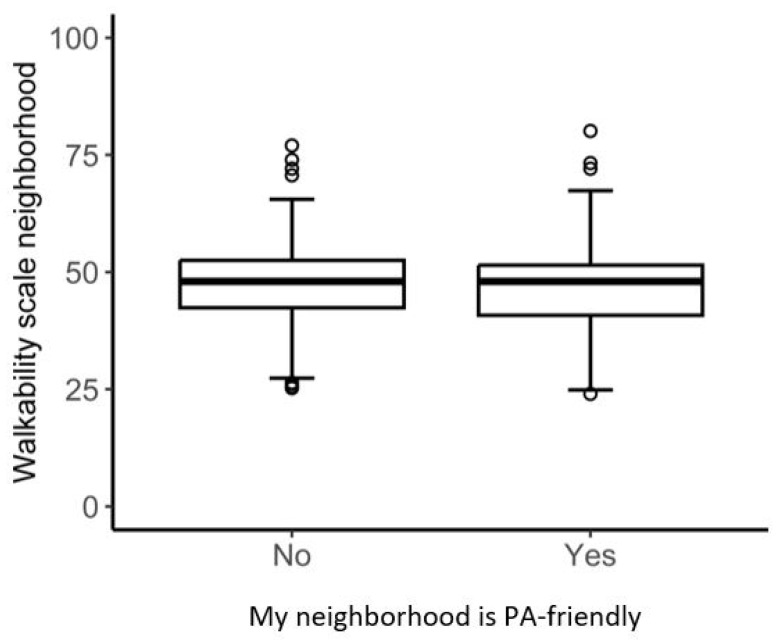
The variability of the walkability index for the two levels of perception.

**Figure 3 ijerph-22-00536-f003:**
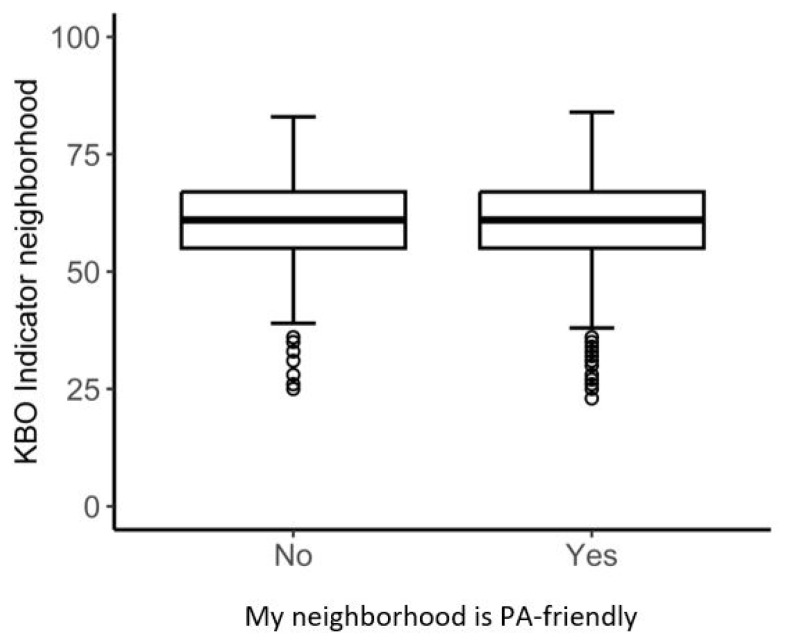
The variability of the KBO indicator for the two levels of perception.

**Table 1 ijerph-22-00536-t001:** Descriptive characteristics of respondents round six (2013–2017).

	Total Study Population Round Six (*n* = 3438)
Sex, Female (%)	53.2
Age	
40–49	7.0
50–59	29.6
60–69	37.6
70–79	21.3
80–89	4.5
Education (%)	
Low	43.5
Middle	30.6
High	25.9
Working, yes (%)	42.5
Physical functioning (%)	
0–25 points	3.6
25–50 points	5.7
50–75 points	15.9
75–100 points	74.8
Are there conditions in your neighborhood that make it less appealing to engage in sports and physical activity?	
No (%)	83.6

## Data Availability

Walkability data are available upon request via the GECCO website (www.gecco.nl). KBO-indicator data for a PA-friendly environment is publicly available at https://statline.rivm.nl/#/RIVM/nl/dataset/50121NED/table?ts=1719266683682, access date 5 June 2024. The data of the Doetinchem Cohort Study cannot be placed in a public repository due to legal and ethical constraints. The participants’ informed consent did not include consent to the public availability of the data. However, the data are available upon reasonable request by contacting the scientific committee of the Doetinchem Cohort Study by email: doetinchemstudie@rivm.nl.

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
