# Peer review of "Physical Activity Friendliness of Neighborhoods: Do Subjective and Objective Measures Correspond Within a Mid-Sized Dutch Town?"

_ijerph, 2025, doi:10.3390/ijerph22040536_

Round 1
Reviewer 1 Report
Comments and Suggestions for Authors
I thank the authors for an interesting paper on a relatively new construct in assessing physical activity concepts. The paper is well written and the analyses conducted relevant. I just had a few comments/ suggestions and clarifications. On line 108 the authors looked at work status in two categories. Employed or not employed. The sample however based on the age ranges 40-89 should include a fair amount of retirees. Were they classified as unemployed? if so would that skew the demographic data?
Line 124 - Raster cells which i understand to be a grid would benefit from an operational definition. was the grid a subset of the buffer zone . Is each cell the 25x25m grid. that was not very clear.
For the variable perception a single item measure is noted in the descriptor. Would it have been useful to use other aspects of the neighbourhood to tease out the varying parameters involved in the assessment. It cites less appealing or what would make the neighbourhood more appealing. This is very subjective and there may be a fine line of demarcation between appealing at one level of the neighbourhood environment vs the other. is there a more objective measure of appeal that could be used in terms of outlining key parameters that would be useful to all participants?
Table 1 shows that mainly older persons reported high physical functioning is this a cultural phenomenon? This is unlike other developed countries where PA declines with age.
Reviewer 2 Report
Comments and Suggestions for Authors
I hope this letter finds you well. I had the opportunity to review your article titled, “Physical activity friendliness of neighborhoods: do subjective measure correspond within a mid-sized Dutch town?”, which was submitted International Journal of Environmental Research and Public Health.
- Introduction
-. This study approaches the importance of physical activity with the theoretical background of an ecological model.
-. Citing support from the WHO and the Dutch government that an eco-friendly environment can promote physical activity, the study emphasizes the need for research.
-. In addition, this study is considered to be original compared to previous studies in that it deals with medium-sized cities while the basic research was focused on large cities, and includes different measurement methods and various age groups.
-. And the new research design that complements the limitations of existing research was very impressive.
-. However, this study fails to present a hypothesis that differentiates it from existing studies.
- I am concerned about the reliability of the results because the subjective indicators for evaluating the PA-friendly environment are unclear.
-. Although this study notes that previous studies have reported low correlations between objective and subjective indicators, it is unclear whether it seeks to test a new hypothesis.
-. Lastly, if more specific discussions are included on the causes of discrepancies suggested by existing studies, the necessity of the research is expected to be further emphasized.
- Method
-. It is believed that the research method of this study used large samples and reliable data.
-. The reliability of the research results was increased by approaching the physical activity-friendly environment by integrating objective and subjective indicators.
-. In addition, we applied multi-level statistical analysis such as logistic regression analysis to consider clustering effects among participants in the same region, and analyzed interactions to evaluate moderation for socio-demographic variables.
-. However, it is very regrettable that the physical activity-friendly environment was evaluated using a dichotomous questionnaire.
-. The conceptual difference between objective environmental indicators and subjective experiences was not reflected. It would be better to present this part as a limitation of the study.
-. Due to regional specificities, it is difficult to generalize to the entire Netherlands.
- Results
-. The results of this study were able to clearly analyze the relationship between objective data and subjective perception and empirically confirm it in the case of a medium-sized city in the Netherlands.
-. When assessing walkability and physical activity friendliness, it is well organized to consider the influence of the spatial extent of the objective environment on the results by setting the distance radius in various ways.
-. The results of this study are considered excellent in that they provide an opportunity to confirm differences according to city size.
-. However, the measurement method of subjective perception was overly simplified and failed to reflect subtle differences.
- There are limitations in determining whether the objective environment influences subjective perception or whether subjective perception influences environmental evaluation.
- Discussion
-. The discussion showed the researcher's efforts to solve the purpose of the study.
-. The researcher reaffirmed the previous studies reporting low correlations between objective and subjective environmental perceptions, and interpreted the results in a broader context.
-. Presents various factors that may influence subjective perception.
- It is emphasized that in terms of policy, not only objective environmental indicators but also the subjective perceptions of residents should be taken into consideration.
-. It is expected that future research will also be helpful in subsequent studies as it suggests the need to deeply analyze the differences between objective and subjective environmental assessments through qualitative research.
- Conclusion
-. The conclusion summarizes the contents of the study and suggests the academic and empirical applicability of this study.
Reviewer 3 Report
Comments and Suggestions for Authors
Dear Authors,
First, I would like to commend you on your efforts in conducting this important study. The manuscript presents a well-designed and clearly articulated investigation into the relationship between two Dutch geospatial measures and residents' perceptions regarding the physical activity-friendliness of their neighborhoods. The study provides valuable insights, but there are several areas that would benefit from further clarification or revision. To ensure the manuscript meets the standards for publication, I kindly request that you address the following specific comments:
Abstract
- Lines 28-30: The conclusion section could be strengthened. Please provide a more solid summary of your key findings.
- Keywords: Rephrase for clarity and precision. Avoid vague terms such as “physical activity friendliness of neighborhoods.”
Introduction
- The introduction sufficiently explains the topic. However, more information on the geospatial measures would be beneficial. Additionally, I’ve included a few minor comments below:
- Line 43: The term "PA" has been used earlier in the text. Please double-check for consistency.
- Lines 55-58: A citation is needed for this section.
- Line 74: Consider elaborating on why the Netherlands is a particularly relevant setting for this research.
- Line 83: It would be helpful to include the study’s hypotheses here.
Materials and Methods
- This section is well-organized. Great job!
Results
- The results section, including tables and figures, is generally well-presented. I appreciate the use of subheadings, which enhances readability for the audience.
Discussion
- The discussion is well-structured overall, though there are a few areas that require attention:
- Line 265: The term “other studies” is vague. Please specify which studies you are referring to.
- Line 272: Please ensure the reference style is consistent and correct.
- Line 284: When you refer to “most studies,” please specify which ones.
- Line 338: The phrase “Previous European studies” is too broad. Please cite specific studies and avoid general terms like this throughout the manuscript.
